# Phylogenetic and Morphological Analysis of Wing Base Articulation in Vespidae (Hymenoptera): A Cladistic Approach

**DOI:** 10.3390/insects17010039

**Published:** 2025-12-27

**Authors:** Hasin Ullah, Xiaojuan Huang, Yao Zhang, Jia Li, Danyang Zhu, Chenlu Yang, Yuan Hua, Lian-Xi Xing, Jiangli Tan

**Affiliations:** Shaanxi Key Laboratory for Animal Conservation/Key Laboratory of Resource Biology and Biotechnology in Western China, Ministry of Education, College of Life Sciences, Northwest University, Xi’an 710069, China; hasenullah888@yahoo.com (H.U.); huangxiaojuan@stumail.nwu.edu.cn (X.H.); zhangyao97321@163.com (Y.Z.); lejea2020@163.com (J.L.); zhudanyang@stumail.nwu.edu.cn (D.Z.); 18792407938@163.com (C.Y.); hy@nwu.edu.cn (Y.H.); lxxing@nwu.edu.cn (L.-X.X.)

**Keywords:** cladogram, micro-CT, TNT, PAUP, phylogeny

## Abstract

The study identifies previously overlooked variation in wing base sclerites among vespid wasps and explains how these structures support folding, control, and stability of the wings. Distinct features of the first, second, and third axillary sclerites, along with the basiradial bridge, show clear functional and evolutionary patterns within the family. Forewing and hindwing base characters were coded for sixteen vespid taxa and *Xyela* sp., but the resulting topology does not recover Vespidae as monophyletic under the present morphological dataset. The work also corrects earlier ideas regarding the presence of a fourth axillary sclerite. It introduces detailed articulation models and muscle sketches that provide a firmer basis for future biomechanical and comparative studies.

## 1. Introduction

The base of the wing and its membrane create a complex arrangement essential for wasps’ flight capabilities. This structure consists of several interdependent components, including the axillary sclerites and their associated muscles. The axillary sclerites typically comprise three main elements. They share a common structural blueprint [1]. The first axillary sclerite (1Ax) is a longitudinal element that consists of the head, neck, and basal sections. It forms an α angle between its proximal and distal segments [2], and articulates with the anterior notal wing process (ANWP). The second axillary sclerite (2Ax) is triangular and has a broad projection on its ventral side that extends beneath and connects to the 1Ax. The third axillary sclerite (3Ax) is also longitudinal, positioned between the proximal notal wing process of the notum (PNWP) and the wing membrane [3]. The third axillary sclerite shows structural modifications across Vespidae, but the degree and form of these changes differ among species, as described in earlier comparative accounts [4]. These axillary sclerites are interconnected by a network of delicate, nearly translucent membranes, which allow insects to fly, steer, and fold their wings during rest.

Wasps of the family Vespidae were chosen as the focal group for this study due to their diverse morphological and ecological characteristics. Vespidae is a large and diverse family comprising over 5210 species [5,6,7]. These wasps exhibit various adaptations, including differences in body size, coloration, nesting behaviors, and social organization, which range from solitary nesting—where individual females construct and provision nests independently—to fully eusocial colonies with cooperative brood care. Their wing articulation is crucial for flight performance, influencing foraging efficiency, predatory behavior, and social interactions. Given the structural complexity of the wing base, analyzing its components provides an opportunity to explore evolutionary relationships among different vespid subfamilies. Recent molecular analyses [8,9] have provided well-supported hypotheses for major vespid lineages. These studies place Stenogastrinae as sister to the Polistinae + Vespinae clade. While certain internal relationships within Polistinae remain debated, the higher-level structure of Vespidae is relatively stable across molecular analyses [10,11,12]. In particular, the placement of Stenogastrinae and the internal structure of Polistinae remain contentious, and morphological characters from the wing base have rarely been integrated into these analyses. Recent studies have suggested that the absence of a fourth axillary sclerite in Vespidae may challenge previous assumptions about wing base composition in Hymenoptera, necessitating a reevaluation of axillary homology and function.

This study incorporates a forewing base model and a hindwing base model, detailing the first, second, and third axillary sclerites. Additionally, we present high-resolution images of the wing base from multiple vespid species, capturing interspecific variations in articulation structures. These morphological characters were analyzed to explore structural variation in the wing base within selected vespid lineages. Because the taxon sampling is limited, the dataset cannot formally test subfamily monophyly. However, by integrating wing-base morphology with phylogenetic analysis, this study refines our understanding of vespid subfamily relationships.

The complexity of wing articulation in insects has been thoroughly described and analyzed in various studies, with particular emphasis on the significance of axillary sclerites in supporting wing mobility and flight stability. Snodgrass’s foundational work has established a wing articulation model that typically features three axillary sclerites (1Ax, 2Ax, and 3Ax). Notably, certain taxa, such as Hymenoptera and Orthoptera, exhibit an additional fourth axillary sclerite (4Ax), as demonstrated by [13,14]. This increased structural complexity can be linked to adaptive radiation and specialized flight mechanics within these groups [15,16]. The anatomical arrangement surrounding the axillary sclerites plays a crucial role in the functionality of wing movement. The tegula acts as a bridge connecting the wing base and thorax, and it is enriched with sensory hairs that are essential for detecting airflow, thereby enhancing flight stability and coordination during wing movement [15,17]. The articulation of the axillary sclerites with the notum relies on the interaction between the anterior notal wing process (ANWP), median notal wing process (MNWP), and posterior notal wing process (PNWP). Specifically, the first axillary sclerite articulates with the ANWP and MNWP, while the third connects to the PNWP, facilitating coordinated movement essential for flight [18,19]. Additionally, the structural relationship between the proximal and distal median plates and the wing veins, including the media (M), cubitus (Cu), and cubitus posterior (PCu), illustrates a sophisticated design for aerodynamic efficiency [20]. Dynamic interactions among these anatomical features have been underscored in phylogenetic studies, which highlight the conservation of wing articulation components across various insect orders and provide critical insights into evolutionary relationships and functional morphology [3]. Hörnschemeyer emphasizes that conserved features at the genus and family levels serve as valuable phylogenetic characters, reinforcing the idea that axillary structure is pivotal in the evolutionary tree of insects [2]. Moreover, our work elucidates the basiradial bridge, which is speculated to be a remnant of the second axillary sclerite (2Ax) and likely augments the overall stability and functionality of the wing articulation framework [16]. Furthermore, the uniqueness of wing base structures among different taxa suggests that these characteristics are evolutionarily significant and influence both flight mechanics and the behavioral ecology of the species [21]. The relationships identified in these studies are instructive for understanding how morphological traits can reflect phylogenetic affiliations and guide future research in evolutionary biology and entomology [21].

Our current study documents morphological variation in the wing base of Vespidae and applies these data to unresolved ambiguities within the family. By integrating detailed analyses of forewing and hindwing articulation sclerites into a phylogenetic framework, it seeks to clarify long-standing ambiguities in vespid relationships and to evaluate the evolutionary significance of wing articulation structures. The incorporation of wing base morphology into phylogenetic analyses provides new perspectives on the evolution of wing form and biomechanics and offers insights into broader evolutionary patterns, including behavioral and social diversification in wasps.

## 2. Materials and Methods

### 2.1. Specimen Collection and Preservation

Sixteen vespid species were collected using a sweep insect net with a 38 cm diameter opening, following standard entomological methods. The net was constructed of fine nylon (mesh size approximately 0.5–1.0 mm) mounted on a steel ring with a wooden handle [22,23,24]. Flying wasps were captured by aerial sweeping, and immediately after capture, the net was twisted to close the opening, allowing insects to settle at the bottom of the net. Specimens were immobilized using an ethyl acetate killing jar and transferred to ventilated vials. During shipment to the laboratory, specimens were protected from direct sunlight and excessive heat using insulated containers. In the laboratory, specimens were preserved in 70–80% ethanol, which was replaced after 24–48 h to remove residual water [25,26]. Each specimen was labeled with complete collection data. The voucher specimens were preserved in our research lab, College of Life Science, Northwest University, Xi’an, Shaanxi, China (NWUX). The complete details are provided in Table 1. The axillary sclerites were removed from the body and observed in glycerin under the ultra-depth three-dimensional microscope system VHX-5000 (Keyence Corporation, Osaka, Japan). All drawings were made according to the photos of the right wings, with the orientation of the axillaries described with the body’s main axis. This perspective was chosen to present interspecies differences most comprehensively. Additionally, PowerPoint 2010 was used to outline each observed fragment, and Adobe Photoshop 2020 was used to edit the images.

### 2.2. Micro-CT Imaging and 3D Reconstruction

Micro-computed tomography (micro-CT) was used to obtain high-resolution scans of the thoracic region, enabling the visualization of the internal arrangement of the wing base structures. The reconstructed image stacks were imported into the 3D Slicer software (version 5.8.1) for segmentation and three-dimensional reconstruction. This process allowed precise identification and organization of the indirect flight muscles, including the dorsoventral (DVM) and dorsolongitudinal (DLM) muscles, as well as the axillary sclerites and their associated muscles. The resulting 3D models provided accurate spatial relationships among these components, facilitating detailed morphological interpretation and comparative analysis across taxa.

### 2.3. Terminologies

The research of Brodsky [27] and Ninomiya and Yoshizawa [20] explains the nomenclature for wing base sclerites, whereas Wootton’s work [28] provides the basis for concepts of wing articulation. In the text and figures, several abbreviations are employed, including ANWP (anterior notal wing process), 1Ax, 2Ax, 3Ax (axillary sclerites 1, 2, 3), br (basiradiale), brb (basiradial bridge), bsc (basisubcostale), hp (humeral plate), MNWP (median notal wing process), PNWP (posterior notal wing process), and tg (tegula).

### 2.4. Phylogenetic Analysis

As described in the character description in the Results section, 42 characters from the bases of the forewings and hindwings were systematically coded for analysis, with coding focusing only on quantitative aspects when variations were identifiable and not part of a continuous spectrum. Although most groups exhibit similar modifications in the fore- and hindwing base structures [14,16,20]. In this study, the forewing and hindwing show distinct structures, so we selected both bases for data collection.

Each species was regarded as a terminal taxon for phylogenetic analysis. We used TNT Version 1.6 [29] and NONA version 2.0 [30] to look at 14 characters from the hindwing bases and 28 from the forewing bases. Using 100 replications and heuristic parsimony techniques, a bootstrap analysis was conducted with 10,000 iterations, collapsing branches with values of 50% or less. WinClada version 1.00.08 [31] and TNT version 1.6 were used to calculate Bremer’s decay indices. Appendix A contains comprehensive details on the data matrix. The dataset was further examined in PAUP*4.0a10 [32] to test, validate, and compute the CI and RI of every character. This research used the heuristic parsimony strategy, which involved considering characters as unordered and equally weighted, using ACCTRAN optimization, the MulTrees option, and adding 100 random taxa and TBR branches.

## 3. Results

### 3.1. Morphological Comparison of Wing Base Structures

Our observations indicate differences between these regions, and we also included the tail portion in the 1Ax. The third axillary is particularly vital in all the Vespidae species, as it contains a muscle regarded as a defining characteristic. To enhance the clarity of character descriptions, schematic illustrations of the axillary sclerites are provided, though they do not depict the interrelationships among the axillaries. Each sclerite at the wing base is identified by its proximal, distal, dorsal, and ventral orientations. Figure 1 presents the forewing sketch, while Figure 2 illustrates the first-ever sketch of the hindwing base.

### 3.2. General Connection Pattern and Shape of Axillary Sclerites for Fore Wing

The first axillary sclerite (1Ax) articulates with the median notal wing process, serving as a crucial structural component in wing articulation. The first axillary sclerite (1Ax) is connected to the 2Ax through its main body, while its head region articulates with the (bsc) of the wing. The shape of 1Ax is elongated and resembles the English letter “E.” It is divided into the head, neck, body, and tail. A specialized sclerotized structure is dorsoventrally located between the body and tail. The knob is uniquely designed so that when the wing is in the resting position, it does not hit the mesothorax. However, when the wing expands, it comes into contact with the mesothorax. This structure is considered crucial for the wing’s up-and-down motion and digit eight (8) rotation during the flight. The knob functions as a paddle, with the mesothorax striking it to facilitate wing movement.

The 2Ax forms three distinct articulations: proximally with the body of the 1Ax, ventrally with the 3Ax, and distally with the basiradiale. It is triangular, but one arm is elongated. Our study identifies that the 3Ax of vespids exhibits varying degrees of fusion with its distal portion, forming a joint-like structure in some species. This structure articulates proximally with the posterior notal wing process and distally with the basiradiale (ba), while also creating a ventral attachment to the 2Ax. The shape of the 3Ax varies but is typically dorsally rectangular and posteriorly triangular. When stretched from different positions, it increases vibration and enables the wing to move faster, which may significantly influence wing flexibility and articulation. Additionally, the shoulder sclerite is proximally connected to the anterior notal wing process (ANWP) and distally to the humeral plate via a robust membranous attachment, contributing to the overall structural integrity and flexibility of the wing articulation system.

### 3.3. General Connection Pattern and Shape of Axillary Sclerites for the Hind Wing

The connection pattern of the hind wing closely resembles that of the forewing; however, significant morphological modifications have occurred. One notable adaptation is the lock-and-key mechanism between the 1Ax and the 2Ax, where the 1Ax functions as the lock and the 2Ax as the key. During the expansion of the wing, the 2Ax inserts itself into the 1Ax and thus provides structural stability and produces the wing vibration. In contrast, the 2Ax in the forewing functions primarily as an elasticity-generating structure. Unlike in the forewing, the 2Ax and 3Ax are not connected ventrally in the hindwing. Additionally, the morphology of all axillary sclerites exhibits notable differences between the two wing pairs, except for the shoulder sclerite, which remains structurally conserved. The position of the median plate has also undergone modification. The forewing median plate is positioned horizontally relative to the wing venation, whereas the hindwing median plate is positioned vertically. Its size is also reduced to one-fifth of the forewing median plate.

The shape of the shoulder sclerite is triangular. The 1Ax is similar to the Arabic character ‘Hamza’, but it has a ventral attachment to the basisubcostale. The 2Ax is twisted unusually, but does not have a prominent connection with the 3Ax. The 3Ax is bifurcated distally and narrower proximally.

### 3.4. Wing Articulation Mechanism

When the pronator muscle of the shoulder sclerite contracts, the wing expands, while the simultaneous relaxation of the flexor muscle facilitates this movement. As this occurs, the knob-like projection of the first axillary sclerite (1Ax) makes contact with the mesoscutum, initiating wing articulation. The coordinated action of the dorsal longitudinal and ventral muscles further regulates wing motion: relaxation of the dorsal longitudinal muscle causes the wing to lower. In contrast, relaxation of the ventral muscle raises it, operating reciprocally.

A notable structural variation exists in the musculature: the pronator muscle is singular, whereas the flexor muscle is triplicated. This distinction plays a critical role in wing control. While a single muscle is sufficient for basic expansion and relaxation, the triplicate flexor muscles attached to the third axillary sclerite (3Ax) are essential for fine-tuned steering and modulation of wing vibrations during flight. Their function is reinforced by the connection between the second (2Ax) and third (3Ax) axillary sclerites, as well as morphological variations in the 3Ax and basanale, which further optimize wing maneuverability and stability. The 3D reconstruction is shown in Figure 3. Appendix A provides a three-dimensional rotational visualization of the structure.

### 3.5. Vespinae

We studied eight members of the subfamily Vespinae: *Vespula germanica*, *Provespa barthelemyi*, *Dolichovespula flora*, *Vespa basalis*, *V. velutina*, *V. mandarinia*, *V. analis*, and *V. ducalis.* The wing base structure of Vespinae comprises essential articulation sclerites, along with associated fold and flexion lines that largely retain their plesiomorphic condition. However, compared to other subfamilies of Vespidae, some distinct modifications are observed. Some important differences are listed here. The tegula is almost rounded and egg-shaped (Cha:1). The sensilla on the tegula are more (Cha: 4). The shape of the shoulder sclerite is sickle-shaped (Cha: 5). The first axillary sclerite head has an inward groove at the anterior side (Cha: 9). Pseudo-MNWP is negligible (Cha: 13). 3Ax has a ventral outgrowth (Cha: 18). Basisubcostale attachment to 2Ax is weak (Cha: 23). Median plate is wider (Cha: 24). The shape of the first axillary of hindwing looks like the Arabic word ‘Hamza’ and the body is broader long way curving (Cha: 30). The tail portion of the first axillary is narrower (Cha: 33). The most important distinguishable character is the length of the 1Ax sclerite and the more projected node on the 1Ax sclerite body (Cha: 11, 12) shown in Figure 4A,B,C and Figure 6B,C and the hind wing picture and sketches are shown in Figures 7C,D and 8B,C.

### 3.6. Polistinae

The wing base structure of Polistinae consists of essential articulation sclerites, along with associated fold and flexion lines that largely maintain their plesiomorphic condition. However, compared to other subfamilies of Vespidae, certain distinct modifications are evident. Our study focuses on six species within this subfamily: *Polistes jokahamae*, *P. rothneyi*, *P. megei*, *P. tenuispunctia*, *P. snelleni*, and *Parapolybia nodosa.* The shape of the tegula is smooth and more concave not tapering at the ends (Cha: 1). Shape of the shoulder sclerite is tapering at both ends (Cha: 5). Pseudo-MNWP is longer and prominent (Cha: 13). Median plate is more rectangular (Cha: 25). The shape of the head of the first axillary and the body makes a mouth appearance in case of Polistinae is more rounded (Cha: 31). The third axillary sclerite of the hindwing is elongated and bears lobes. In Polistinae, it is slightly straighter (Cha: 39), as shown in Figure 5A,B,C and Figure 6A,D. The hind wing is shown in Figure 7A,B and Figure 8A.

**Figure 4 insects-17-00039-f004:**
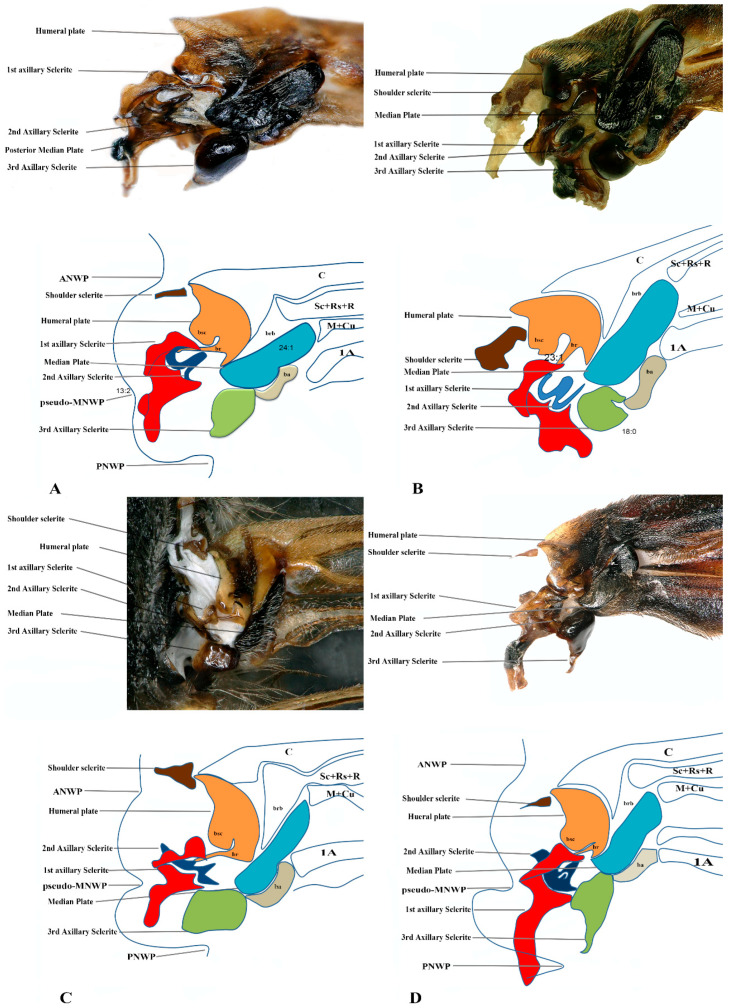
Forewing base articulation sclerites original picture and sketch together; (**A**) *Vespa mandarinia*, (**B**) *Vespa velutina*, (**C**) *Vespula germanica*, (**D**) *Oreumenes decoratus*.

**Figure 5 insects-17-00039-f005:**
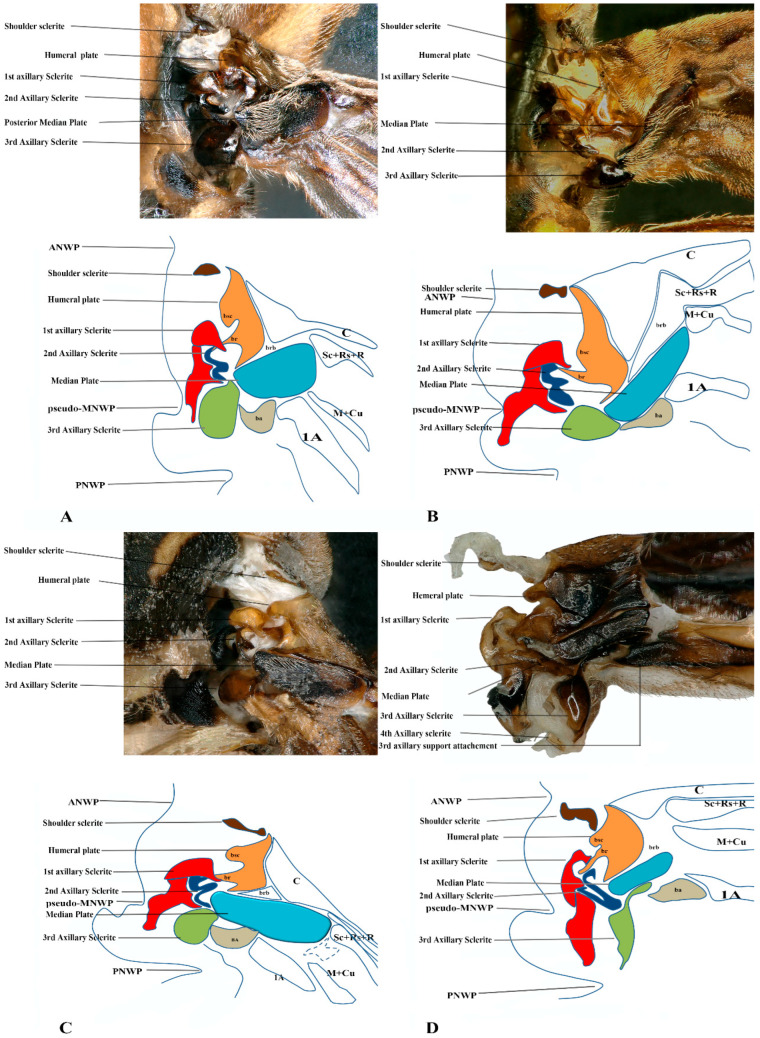
Forewing base articulation sclerites original picture and sketch together; (**A**) *Polistes jokahamae*, (**B**) *P. rothneyi*, (**C**) *P. tenuispunctia* Kim, (**D**) *Eustenogaster nigra*.

**Figure 6 insects-17-00039-f006:**
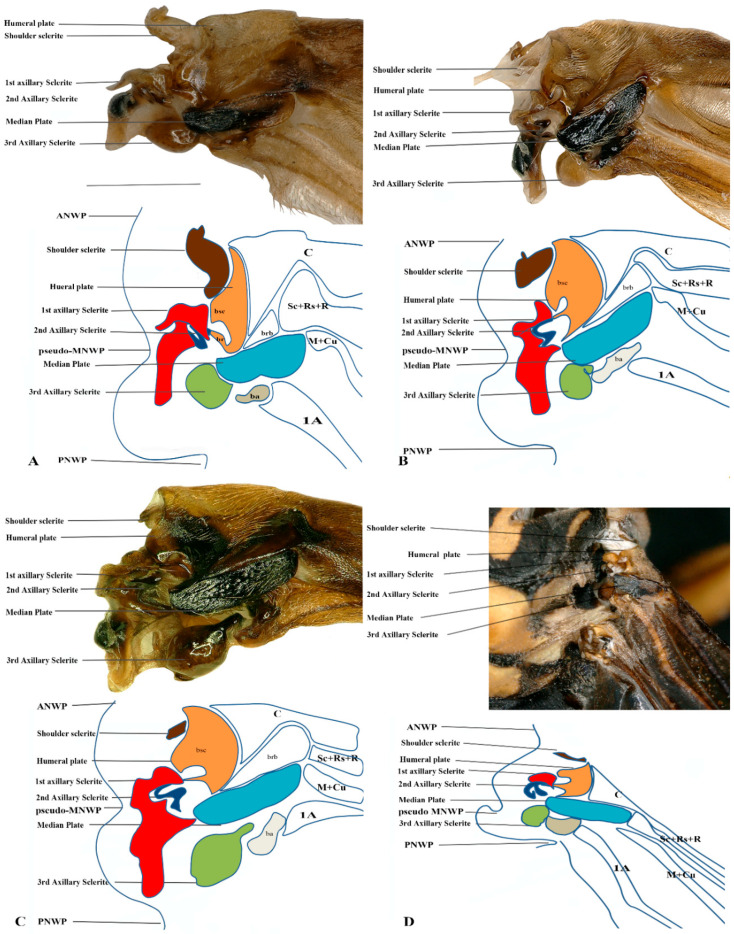
Forewing base articulation sclerites original picture and sketch together; (**A**) *Parapolybia nodosa*, (**B**) *Provespa barthelemyi*, (**C**) *Dolichovespula flora*, (**D**) *Polistes megei*.

### 3.7. Eumeninae and Stenogastrinae

We studied one species from each Eumeninae and Stenogastrinae. While most of the articulation sclerites are the same, we also observed some distinct variations. The surface of the tegula is smooth, slightly concave, and tapered at both ends (Cha: 1). The shape of the first axillary sclerite head is more pointed and curved in *O. decoratus* while tapering at both ends in *E. nigra* (Cha: 9). Second axillary sclerite shape *E. nigra* is triangular (Cha: 14). First axillary neck is thicker and wider (Cha: 10). First axillary node (paddle) is less sclerotized and minimally projecting outward (Cha: 11). The 3rd axillary sclerite is slightly hollow inside (Cha: 20) shown in Figure 4D and Figure 5D.

**Figure 7 insects-17-00039-f007:**
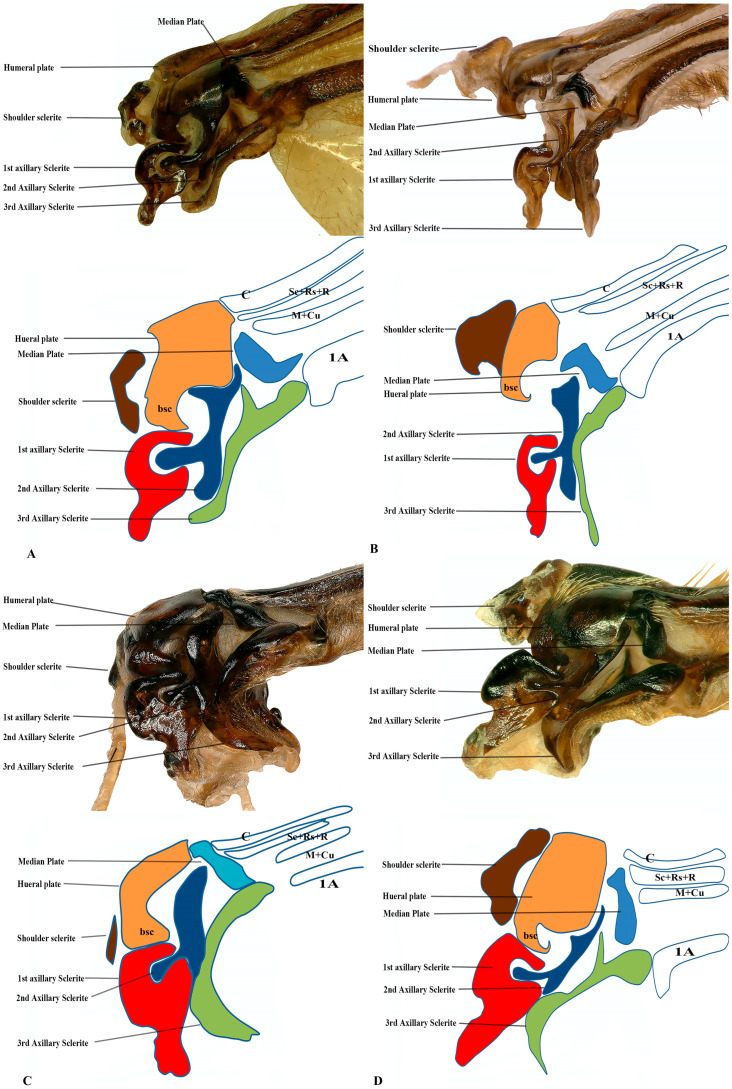
Hindwing base articulation sclerites original picture and sketch together; (**A**) *Polistes jokahamae*, (**B**) *P. rothneyi*, (**C**) *Vespa mandarinia*, (**D**) *V. velutina*.

**Figure 8 insects-17-00039-f008:**
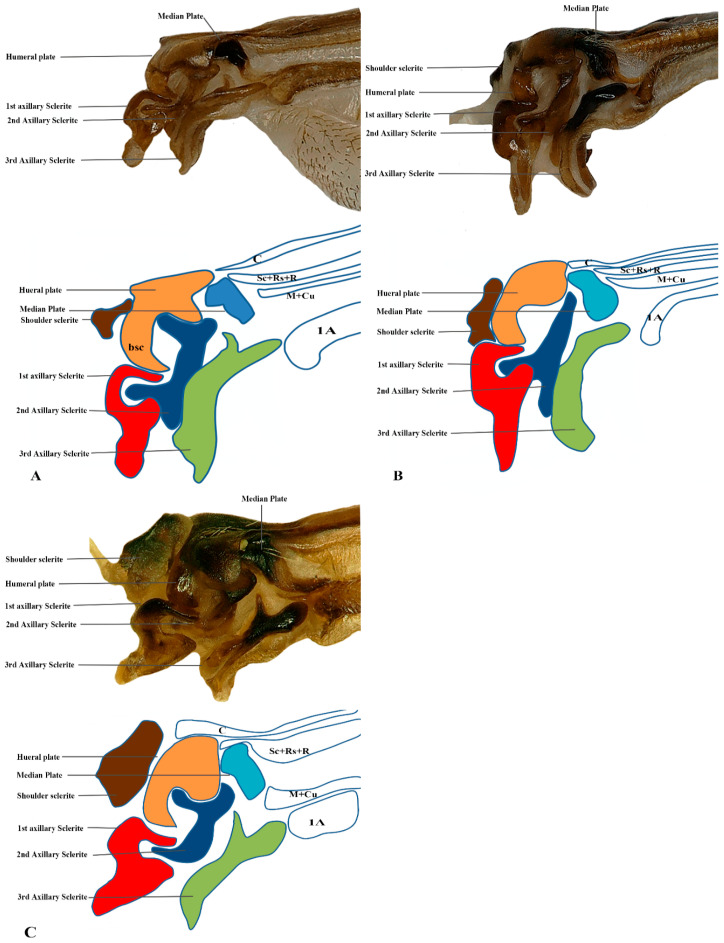
Hindwing base articulation sclerites original picture and sketch together; (**A**) *Parapolybia nodosa*, (**B**) *Provespa barthelemyi*, (**C**) *Dolichovespula flora*.

### 3.8. Character Description of Wing Base Structures for Phylogenetic Analysis

#### 3.8.1. Forewing

Shape of tegula: (0) Smooth and slightly concave and tapered at both ends; (1) Smooth and more concave, not tapering at the ends; (2) Smooth, slightly concave, and tapered at one end; (3) almost rounded or egg-shaped.Tegula covering: (0) Covering half of the median plate, and (1) covering 70 percent of the median plate.Size of the tegula: (0) Relatively small and (1) relatively large.Sensilla of the tegula: (0) Different in size and more sensilla are present; (1) Different in size but few sensilla are present; (2) Very few or absent.Shape of the shoulder sclerite: (0) Tapering at both ends; (1) Tapered at one end and trihedral; (2) Sickle-shaped sclerite.Muscle attachment of shoulder sclerite; (0) Strong attachment at the thorax; (1) weak muscle attachment.Scares around the wing margin: (0) Have scares around the wing margin; (1) Don’t have scares around the wing margin.Pits on the dorsal side: (1) having two pits on the dorsal margin of the fore and hind wings; (2) don’t have the pits on the dorsal margin.First axillary sclerite head: (0) more pointed and curved; (1) tapering at both ends; (2) having an inward groove at the anterior side;First axillary neck: (0) The neck is thinner and wider; (1) Thicker and wider.First axillary node (paddle): (0) Rounded, more sclerotized, and projecting outward; (1) Less sclerotized and minimally projecting outward;First axillary tail: (0) Having a longer tail; (1) Having a shorter tail.Pseudo-MNWP: (0) Having a longer pseudo-MNWP; (1) Having a shorter pseudo-MNWP; (2) Negligible pseudo-MNWP.Second axillary sclerite shape: (0) triangular; (1) A little V-shaped.Second axillary attachment: (0) second axillary attached with brb through a strongly sclerotized strip called br; and (1) through a thinner br.brb and 2nd axillary sclerite size: (0) brb is twice as wide as 2nd axillary sclerite; (1) brb is less wide than 2nd axillary sclerite.Shape of the 3rd Axillary: (0) triangular; and (1) rectangular.3rd axillary: (0) Having a ventral outgrowth; (1) Don’t have a ventral outgrowth;3rd axillary is (0) Flattened; (1) Swollen and raised.3rd axillary sclerite; (0) hollower inside; (1) slightly hollow inside.Attachment of the 3rd Axillary: (0) Attached directly to the ba; (1) Attached indirectly to the BA; (2) almost separated, forming a joint appearance.Basisubcostale: (0) is pointed and projecting inwards; (1) not much pointed and not projecting inward; (2) wider Basisubcostale.Basisubcostale attachment: (0) strong to the first axillary sclerite; and (1) weak attachment to the first axillary sclerite.Median plate is (0) wider and (1) thinner.Median plate is (0) more rectangular; (1) nearly rectangular, but one side is more tapered.Posterior inward groove in median plate: (0) Having a groove posteriorly; (1) Don’t have a groove posteriorly.ba: (0) Smaller than 3rd axillary; (1) bigger than 3rd axillary.Shape of the ba: (0) almost the same as 3rd axillary; (1) Different from 3rd axillary.

#### 3.8.2. Hindwing

29.Tegula: A true tegula of the hind wing is absent in all examined species within Vespidae. However, a pseudo tegula is present. This structure resembles the forewing tegula but differs from the forewing; it is referred to as a pseudo tegula because it is not movable like the forewing tegula. (0) It nearly covers half of the articulation sclerites; (1) It covers approximately 70 percent of the articulation sclerites of the hind wing.30.1st Axillary: The shape of the first axillary looks like the Arabic word ‘Hamza’. (0) The body is broader, long way curving; (1) The body is thinner and quickly curved.31.The shape of the head of the first axillary and the body makes a mouth appearance; (0) more rounded; (1) is a little rounded and oblique.32.The size of the first axillary mouth: (0) Relatively small in size; (1) Relatively large.33.The tail portion of the first axillary is (0) wider, and (1) narrower.34.No proper sclerotized node is seen on the tail portion, but (0) a little sclerotized node is seen; (1) No sclerotization is seen.35.Interproximal portion of the first axillary: (0) With projection; (1) without projection.36.Contact between the first axillary and the bsc: (0) Strong strip; (1) weak strip.37.Shape of the second axillary: The shape of the second axillary is irregular. The second axillary has an important outgrowth that fits into the interproximal portion (mouth) of the first axillary. (0) a little thin outgrowth; (1) a little thick and strong outgrowth.38.Connection between second axillary and brb: (0) More prominent; (1) Not more prominent.39.Shape of the 3rd axillary: The 3rd axillary is elongated in shape and has lobes. (0) a little straighter; (0) a little U-shaped.40.3rd axillary and ba connection: It is directly connected to the ba. (0) Near the ba connection area is more sclerotized; (1) less sclerotized.41.Median Plate: The hind wing median plate is parallel to the body. (0) a little rectangular; (1) A little triangular.42.DMP and HP: (0) Separated by basiradial bridge (brb), and the bridge is more prominent; (1) also separated by brb, but the bridge is not prominent.

### 3.9. Wing Base Phylogenetic Relationship 

Our matrix analysis, employing the matrix method, produced a single most parsimonious tree (MPT) with a tree length of 67, a consistency index (CI) of 0.7612, and a homoplasy index (HI) of 0.2388. The CI excluding uninformative characters was 0.7419, and the HI excluding uninformative characters was 0.3134. The retention index (RI) was 0.8954, and the rescaled consistency index (RC) was 0.2581. Since the analysis yielded only one most parsimonious tree, this topology is presented in Figure 9. Additionally, PAUP*4.0b10 confirmed these identical results. The phylogenetic analysis identifies the subfamily Vespinae as a distinct and monophyletic lineage, with *V. germanica* positioned at the root of the tree. Within Vespinae, *P. barthelemyi* and *D. flora* form a sister group, which clusters with a clade of *Vespa* species. Notably, *V. analis* and *V. ducalis* form a group that is sister to a clade composed of *V. mandarinia* and *V. basalis*, with *V. velutina* placed as their nearest relative. These relationships suggest significant morphological cohesion among *Vespa* species and support their close evolutionary history within Vespinae. These patterns support the current classification of the Vespinae and reflect their distinct lineage within the family.

The subfamily Polistinae forms a separate clade from Vespinae, highlighting its evolutionary distinction. Within Polistinae, two major clades are recovered: one includes *P. jokahamae*, *P. megei*, and *P. tenuispuncta* (bootstrap support 52%), while the other includes *P. rothneyi* and *P. snelleni* (bootstrap support 80%). *Parapolybia* is placed independently within the broader Polistinae grouping. These relationships support genus-level divergence within Polistinae and reinforce its separation from Vespinae.

The tree clearly separates the social subfamilies Vespinae and Polistinae from the predominantly solitary Stenogastrinae and Eumeninae. This pattern is not explained by wing-folding behavior, as previously suggested, because wing folding is widespread and evolutionarily complex in Vespidae. Most subfamilies, including Eumeninae sensu lato, Polistinae, Vespinae, and several Masarinae genera, fold their wings, whereas Stenogastrinae, Euparagiinae, and most Masarinae do not, indicating at least two independent origins of this trait. Behavioral syndromes are similarly diverse across subfamilies. Thus, although *Oreumenes* folds its wings while *Eustenogaster* does not, wing folding does not align with the phylogenetic structure recovered here. Instead, the distinct placement of Stenogastrinae and Eumeninae likely reflects deeper morphological differences rather than wing-folding or sociality patterns, as shown in Figure 9.

## 4. Discussion

### 4.1. Axillary Sclerites

Axillary sclerites exhibit significant variations in structure and articulation among the sixteen species examined in our study. The examined species possess three axillary sclerites (1Ax, 2Ax, 3Ax) and a shoulder sclerite, with differences primarily in the shape and articulation of these structures. Snodgrass reported that Hymenopteran species such as *Sirex flavicornis*, *Pepsis* sp., and the honey bee have four axillary sclerites [13], which contrasts with our findings. However, Mikó and Deans documented three axillary sclerites in Hymenoptera based on their examination of *Macroxyela*, *Xyela*, and *Athalia* [33], supporting our observations. The structural composition of the insect’s wing base has been previously outlined and analyzed by [34,35]. Among its various elements, the first and third axillaries hold significant functional and phylogenetic importance [3]. While the second axillary serves as a crucial support for wing movement, the primary transmission of flight muscle forces from the notum to the wing largely depends on the first axillary. Furthermore, our study reveals that the first axillary sclerite (1Ax) in Vespidae is composed of distinct regions: the head, neck, body, and tail. This structure can be categorized into three distinct sections: the head, the neck, and the basal part (body). Ninomiya & Yoshizawa [20] noted that in Holometabola, the head and neck are not distinctly separated. A significant feature we identified is a well-sclerotized and prominent knob located between the tail and body portion. This knob plays a crucial role in wing articulation, enabling the 1Ax to move up and down in coordination with the mesothorax. These findings align with the descriptions of Zhao [4] and Snodgrass [13], but our study expands on their models by adding the presence of the knob and tail, highlighting their functional significance in wing movement.

Our findings also show that the second axillary sclerite (2Ax) in Vespidae is a single, unified structure, differing from the two-part composition observed in *Macroxyela ferruginea*, *Athalia rosae*, and *Xyela* sp. [33]. Whereas our study on Vespidae reveals a single, unified 2Ax structure that aligns with broader Hymenoptera variations, Snodgrass illustrated *Sirex flavicornis*, *Pepsis* sp., and honey bees as having distinct 2Ax morphologies [13]. These differences suggest that axillary sclerite morphology varies across Hymenoptera, potentially reflecting adaptations for specific flight mechanics. In Vespidae, the 2Ax is triangular and forms three attachments, crucial for wing folding and positioning. This structure likely enhances flight efficiency by generating high-frequency wing vibrations as it is pulled from opposing directions.

### 4.2. Third Axillary Sclerite and Basanale

In contrast to previous studies by Yoshizawa and Saigusa [14], who suggested that the distal portion of the third axillary sclerite is an independent structure (the basanale), and the findings of Ouvrard [1], which indicates partial fusion with the third axillary sclerite, our observations reveal a more complex pattern of fusion. In species such as *O. decoratus* and *E. nigra*, the third axillary sclerite and its distal portion exhibit varying degrees of fusion, with some species showing partial fusion and others displaying a strong fusion, creating a joint-like structure. Furthermore, contrary to Yoshizawa and Saigusa [36], given the assumption that the lack of a connection between the third and second axillary sclerites is an artifact, we observed a strong ventral attachment between these two sclerites. A single muscle may suffice for basic wing expansion, but the presence of three flexor muscles attached to 3Ax indicates a more specialized function, particularly in steering and modulating wing vibrations. This is further supported by the connection between 3Ax and 2Ax, which may act as a stabilizing mechanism that allows for precise flight adjustments.

### 4.3. Basiradial Bridge

Our findings contribute to understanding the basiradial bridge in insect wing morphology, offering insights into its potential homology. Described as a sclerotized structure lacking cuticular micro-sculpture, it aligns with Brodsky [27]. The definition of a bridge connecting the subcostal and radial veins reinforces wing stability. However, Yoshizawa & Saigusa [20] suggest it may be the distal fragment of the second axillary, contributing to wing articulation and movement. The basiradiale’s role in wing folding and vein integration supports the idea that the basiradial bridge, or second axillary fragment, functions as part of a coordinated system for wing stability and flexibility.

### 4.4. Phylogeny

The analysis did not recover Vespidae as monophyletic. This outcome likely reflects the limited taxon sampling and the restricted character system, which may not provide sufficient resolution for deeper relationships. Vespinae forms a monophyletic clade with moderate support (62%), while Polistinae, Stenogastrinae, and Eumeninae cluster together with strong support (98%). However, this grouping contradicts all major molecular studies [7,37,38,39], which consistently recover Stenogastrinae as the sister group to the remaining Vespidae, but do not place it as sister to Eumeninae. Our topology, therefore, appears to reflect analytical limitations rather than true evolutionary relationships, and it differs further from earlier morphological studies [10,11], which proposed alternative subfamily arrangements based on broader character sampling.

Our phylogenetic analysis revealed variation in homoplasy among morphological traits. Characters 2–8, 11, 14, 15, 19–22, 25, 26, 29, 31–32, 34–42 (CI = 1.000, RI = 1.000) showed strong synapomorphic value, providing robust phylogenetic signals. In contrast, Characters 23, 24, 30, and 33 (CI = 0.250–0.333) exhibited significant homoplasy, suggesting potential convergent evolution. Characters 1, 9, 10, 12, 13, 16–18, 27, and 28 (CI = 0.500–0.750) exhibited moderate homoplasy, which may have contributed to character conflict and reduced phylogenetic resolution.

The monophyly of Vespinae supports a common evolutionary origin for the highly eusocial condition observed in this subfamily [40,41]. In contrast, the strong support for the Polistinae–Stenogastrinae–Eumeninae clade (98%) in our analysis cannot be interpreted as evidence of a shared “less rigidly social” ancestor, because most Eumeninae are strictly solitary and do not exhibit intermediate forms of sociality. This interpretation also does not reflect Carpenter’s conclusions [10]. Moreover, the placement of Stenogastrinae as sister to Polistinae and Eumeninae in our results does not provide support for the hypothesis that sociality evolved from solitary ancestors [10,42]. Rather, the topology likely reflects limitations in the character set and does not recover the well-established pattern of solitary-to-social transitions documented in broader phylogenetic studies.

Despite some homoplastic traits, the overall Consistency Index (CI = 0.7612) and Retention Index (RI = 0.8954) suggest that the majority of the morphological characters are phylogenetically informative. The Homoplasy Index (HI = 0.2388) indicates that approximately 23% of character-state changes are due to convergent evolution or reversals. Future research should consider refining character coding strategies and incorporating molecular data to improve phylogenetic resolution.

Limitations include the absence of Euparagiinae and Gayellini, as well as Masarinae sensu stricto, Raphiglossinae, Zethinae, and most representatives of Eumeninae. The latter subfamily is particularly diverse, comprising nearly three-quarters of all Vespidae and encompassing several major clades [8,9]. Because these groups are underrepresented or missing from the present dataset, the analysis cannot reliably clarify relationships among the vespid subfamilies. Future research should expand taxon sampling and integrate broader morphological and molecular evidence to establish a more robust phylogenetic framework.

## 5. Conclusions

The complex articulation of the wing base in Vespidae showcases evolutionary adaptations that enhance flight efficiency, maneuverability, and wing folding. The axillary sclerites (1Ax, 2Ax, and 3Ax), along with structures such as the tegula, median plates, and pleural wing process, create a conserved yet functionally diverse framework among species. Our study identifies a sclerotized knob in 1Ax, contributing to wing articulation, while the 2Ax’s triangular shape enhances wing folding and high-frequency vibrations. The fusion patterns in 3Ax influence wing flexibility and stability, and the basiradial bridge integrates the subcostal and radial veins, reinforcing wing mechanics.

This study introduced two models illustrating forewing and hindwing articulation, refining our understanding of Hymenopteran wing base morphology. The present dataset is insufficient to test the monophyly of Vespidae or to clarify the subfamily-level relationship. The absence of a fourth axillary sclerite challenges prior classifications, emphasizing the need for further biomechanical and comparative studies to explore wing adaptations and evolutionary diversification.

## Figures and Tables

**Figure 1 insects-17-00039-f001:**
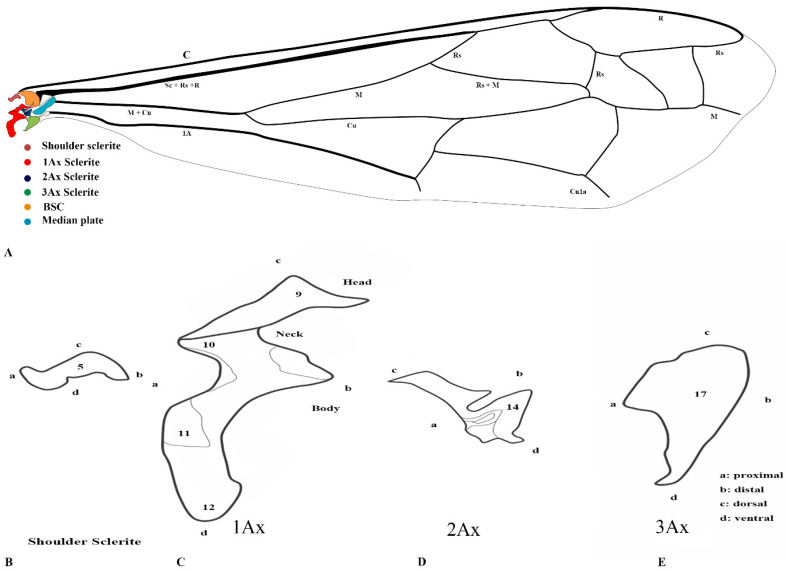
Forewing sclerites model description for the family Vespidae (Hymenoptera); (**A**) Forewing sketch along with colored wing articulation sclerites, (**B**) Shoulder sclerite, (**C**) 1Ax, (**D**) 2Ax, and (**E**) 3Ax. The numbers correspond to specific morphological characteristics, with detailed descriptions available in the character description section.

**Figure 2 insects-17-00039-f002:**
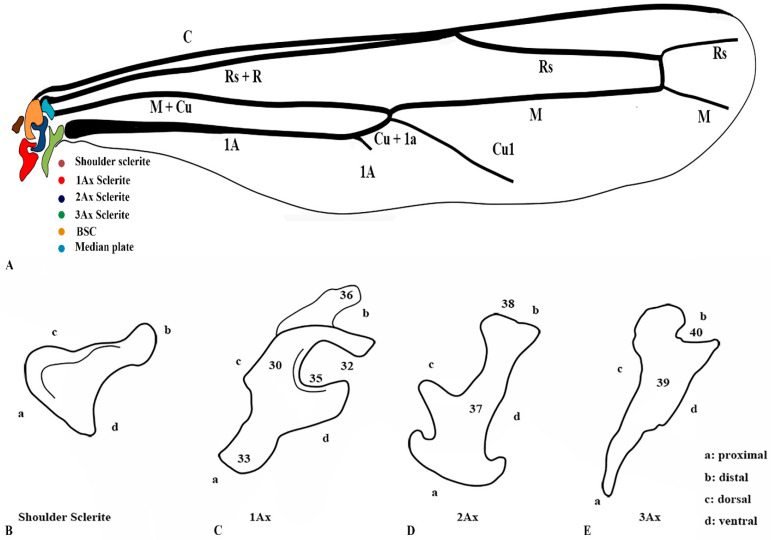
Hindwing sclerites model description for the family Vespidae (Hymenoptera); (**A**) Hind wing sketch along with colored wing articulation sclerites, (**B**) Shoulder sclerite, (**C**) 1Ax, (**D**) 2Ax, and (**E**) 3Ax. This is a newly developed model. The numbers correspond to specific morphological characters, with a detailed description section.

**Figure 3 insects-17-00039-f003:**
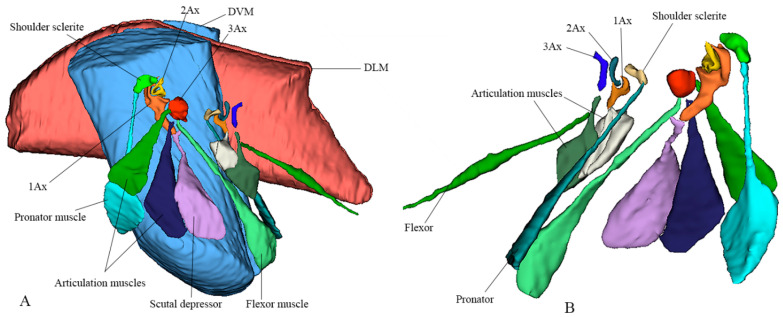
Three-dimensional sketch using micro-CT and 3D Slicer to depict the exact shapes of the indirect muscles (DVM, Dorsoventral Muscle; DLM, Dorsolongitudinal Muscle). (**A**) Lateral view showing labeled forewing articulation sclerites and corresponding muscles. (**B**) Ventral view showing labeled hindwing articulation sclerites and corresponding muscles.

**Figure 9 insects-17-00039-f009:**
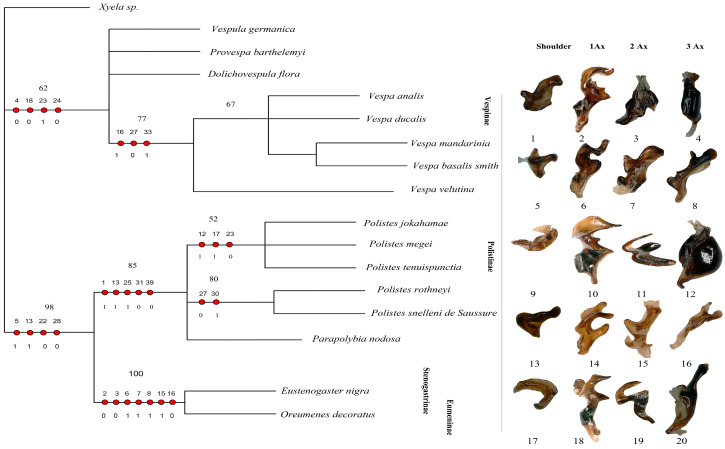
Cladogram of Vespidae based on forewing and hindwing base morphology. The figure presents the single most parsimonious tree (MPT) obtained from the analysis, focusing on the structural characteristics of the forewing and hindwing bases. Unambiguous character mapping is indicated with filled circles for homologous traits and open circles for reversals or parallels. The number above the circle shows the number of characters, and the number below the circle displays the character states. Numbers on the nodes represent bootstrap values and Bremer’s decay indices. The sclerites corresponding to 1Ax, 2Ax, and 3Ax for each subfamily are displayed on the right. Sclerites numbered 1–8 belong to Vespinae; 9–16 belong to Polistinae, and 17–20 belong to Eumeninae and Stenogastrinae. *Xyela* sp. is the outgroup.

**Table 1 insects-17-00039-t001:** Collection localities and sampling periods of the specimens examined.

Specimen	Date	Location
*Vespula germanica*	10/2021	Maowusu Desert, Yulin, Shaanxi, China
*Vespa analis*	08/2021	Du Hui Village, Chang’an, Xi’an, Shaanxi, China
*Vespa velutina*	09/2021	Nanren Village, Chang’an, Xi’an, Shaanxi, China
*Vespa ducalis*	06/2021	Qingyun Village, Jinyun Mountain, Chongqing
*Vespa mandarinia*	08/2022	Huangshan City, Anhui, China
*Vespa basalis*	11/2022	Qianyang County, Baoji, Shaanxi, China
*Polistes jokahamae*	07/2022	Du Hui Village, Chang’an, Xi’an, Shaanxi, China
*Polistes rothneyi*	04/2022	Niejiahe, Chang’an, Xi’an, Shaanxi, China
*Polistes megei*	07/2022	Lidu Village, Chang’an, Xi’an, Shaanxi, China
*Polistes tenuispunctia*	06/2015	Yueba, Foping, Qinling, Shaanxi, China
*Polistes snelleni*	07/2022	Du Hui Village, Chang’an, Xi’an, Shaanxi, China
*Parapolybia nodosa*	06/2023	Nanling, Shaoguan, Guangdong, China
*Eustenogaster nigra*	06/2023	Nanling, Shaoguan, Guangdong, China
*Oreumenes decoratus*	07/2013	Jinsixia, Shangluo, Shaanxi, China
*Provespa barthelemyi*	06/2015	Yueba, Foping, Qinling, Shaanxi, China
*Dolichovespula flora*	08/2023	Xunyangba, Ningshan, Ankang, Shaanxi, China

## Data Availability

All data supporting the findings of this study are contained within the article and the Appendix A.

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
