# Peer review of "Phylogenetic and Morphological Analysis of Wing Base Articulation in Vespidae (Hymenoptera): A Cladistic Approach"

_insects, 2025, doi:10.3390/insects17010039_

Round 1

Reviewer 1 Report

Comments and Suggestions for Authors

The discussion of the results must be rewritten.  The authors wrongly characterize their tree as showing separation of Polistinae and Vespinae from Stenogastrinae and Eumeninae.  In fact, Polistinae are grouped as the sister-group of Stenogastrinae + Eumeninae.

There are also some careless statements about commoness of wing folding, with it characterized as applying to most vespids (true) but then this is contradicted.

Comments on the Quality of English Language

Mostly adequate.

Reviewer 2 Report

Comments and Suggestions for Authors

The manuscript reports interesting original (likely) data that should be published. However, it should be completely rewritten! The interpretation is not clear and sometimes is even not correct. I am not an expert in morphology and cladistics but anyway I have found many errors.

Most important things:

Lines 15–17: “The results strengthen evidence for the monophyly of Vespidae and give a clearer view of relationships among the subfamilies.”

Lines 486–488: “Our study used forewing and hindwing base structural data to infer phylogenetic relationships within Vespidae, supporting its monophyly and well-defined subfamily relationships.”

Lines 525–526: “Phylogenetic analysis supports Vespidae as a monophyletic group and clarifies the subfamily relationships.”

This is not correct. The outgroup is Xyela. By this way any artificial group of Apocrita will be monophyletic. However, even taking this into account, Vespidae is not shown to be monophyletic according to Fig. 9. Thus, there are no evidences of their monophyly. The relationships among subfamilies are also not clearer (see my further comments below).

Lines 75–77: “Previous phylogenetic studies of Vespidae, primarily based on molecular data, have provided important insights but left unresolved questions regarding the relationships among subfamilies and the evolutionary trajectory of morphological traits [7][8][9]. In particular, the placement of Stenogastrinae and the internal structure of Polistinae remain contentious”

Indeed, there are phylogenetic studies of Vespidae primarily based on molecular data (first of all Bank et al. 2017 and Piekarski et al. 2018) but they are not cited in the manuscript! Among the cited references, only 8 contains molecular data, while 9 is even dedicated to Hemiptera, not wasps. I afraid that some AI selected references, didn’t it? Well, the position of Stenogastrinae remains contentious but this is not shown here since both 7 and 8 have the same position of this subfamily (sister to Polistinae + Vespinae).

Lines 86–88: “These morphological data are then analyzed within a phylogenetic framework to address unresolved questions in vespid systematics, specifically the monophyly of subfamilies”

Monophyly of most subfamilies are well resolved except Eumeninae s.l. (with Raphiglossinae and Zethinae) and Masarinae s.l. (with Gayellinae). Just one Oreumenes cannot help in this! At the same time, there is no controversy in monophyly of both Vespinae and Polistinae based on both morphological and molecular data. Therefore, the present study cannot clarify the relationships among subfamilies.

Lines 409–411: “This division may be related to wing-folding mechanisms, as social wasps generally fold their wings longitudinally at rest, unlike many solitary species.”

You don’t know wasp morphology, ethology and systematics. Most Vespidae fold their wings. And most of them are solitary. All Eumeninae s.l. (with Raphiglossinae and Zethinae), Polistinae and Vespinae fold wings, as well as Celonites and Quartinia in Masarinae. Other Masarinae s.l. (with Gayellinae), Euparagiinae and Stenogastrinae don’t fold wings. Thus, the wing folding evolved at least twice in Vespidae. Stenogastrinae are mostly subsocial, Polistinae and Vespinae are social (sometimes social-parasitic) and other subfamilies are mostly solitary (except some subsocial Zethinae and Eumeninae s.str.). So, Oreumenes folds wings, while Eustenogaster doesn’t fold. It is very interesting why the vespid wasps fold their wings and how it is related to their morphology, so it’s a pity that this is not properly discussed in the manuscript.

Lines 435–436: “Mikó and Deans documented three axillary sclerites in Hymenoptera [24]”

It would be useful to indicate certain taxa studied by these authors, may be the number of axillary sclerites varies interspecifically.

Lines 489–491: “Polistinae, Stenogastrinae, and Eumeninae form a strongly supported clade (98%). These results align with molecular studies [30][31][32][33], consistently placing Stenogastrinae as the sister group to other Vespidae.”

This is incorrect. The results are controversial to all other studies, cited here and not cited here (see above), because neither of them place Stenogastrinae sister to Eumeninae!

In fact, the results suggest that the morphology of the wing base articulation has low phylogenetic importance. I can speculate that it is more related to a body shape (“life form”) and size of the wasps in the corresponding taxa. E.g., Oreumenes and Stenogastrinae are with long petiolate metasomal segment 1, while it is shorter in Polistinae and the body is even more robust in Vespinae. It’s a pity that this is not discussed by the authors.

Lines 502–504: “Strong support for the Polistinae-Stenogastrinae-Eumeninae clade (98%) suggests these subfamilies evolved from a less rigidly social ancestor[7]. Stenogastrinae’s placement as sister to Polistinae and Eumeninae supports the hypothesis that sociality evolved from solitary ancestors [7][37].”

Most Eumeninae are truly solitary, they cannot be evolved from a social ancestor, even less rigidly social. I am sure that Carpenter [7] wrote something different. Indeed, the sociality evolved from solitary ancestors, but it is not supported by the present results in any form.

Line 512: “Limitations include the absence of Euparagiinae and Gayellini”

And also Masarinae s.str., Raphiglossinae, Zethinae, and most Eumeninae, because the latter subfamily is very diverse (3/4 of all Vespidae) and contains five major clades (Bank et al. 2017; Piekarski et al. 2018). Therefore, the present study cannot clarify the relationships among subfamilies.

Grammar requires attention. I am not a native English speaker, but some phrases look strange for me. Just some examples:

Line 24: “burrowing into narrow cells.”

Burrowing can be into something loose, like sand or loose ground. Some Vespidae nest in loose ground (Pterocheilus s.l., Quartinia) but this is rather an exception.

Lines 26–28: “Comprehensively examines the axillary sclerites (1Ax, 2Ax, and 3Ax), shoulder sclerite, and associated wing base structures, revealing significant evolutionary adaptations.”

The sentence looks not grammatically correct, please check.

Line 70: “social structures ranging from solitary to eusocial colonies.”

Solitary what? Solitary wasps are not social structures but solitary colonies means that solitary wasp always make nest aggregations? (If so, this is not true).

Line 268: “The sensilla on the tegula are more”

There are more sensillae or the sensilla is larger?

Line 299: “We studied two species from both Eumeninae and Stenogastrinae”

Probably “one species from each Eumeninae and Stenogastrinae”

Line 321: “relatively big”

Relatively large?

Lines 322–323: “Different and more sensilla are present; (1) Different but few sensilla are present”

Different from what?

Lines 326–327: “Muscle attachment of shoulder sclerite; (0) Strong attachment at the thorax; (1) Strong muscle attachment.”

What is the difference?

Line 328: “Scares around the wing margin: (0) Have; (1) Don’t have.”

Have what?

Lines 329–330: “Pits on the dorsal side: (1) having two pits on the dorsal margin of the fore and hind wings; (2) don’t have the pits on the dorsal margin.”

Pits have pits?

Lines 362–363: “This structure resembles the forewing tegula but differs from the hindwing”

So, the pseudotegula differs from the hind wing? What is the sense of this phrase?

By the way, hind is not a combining word.

Line 386: “Separated my brb”

???

Line 437: “the entomological wing base”

What is the entomological wing???

Other uncertainties:

Line 25: “Recently, we have observed”

Looks like rather a reference to a previous work but there is no link.

Line 62: “Our study identifies”

If this is a part of the Results, than it should not be here. If this is a reference to a previous work, the link is missing.

Fig. 9: What is Vespa basalis smith? I guess Smith (starting with capital) is the taxonomic author but most other species are without authors here (except Polistes snelleni).

Minor typos:

Line 303: “Triangular”

Should be “triangular”

Line 427: “The Xyela sp

Should be “Xyela sp.”

Line 449: “But Our”

Should be “but our”

References are very inaccurate. Just one example, no. 34: who is Arthropod, J.C.??? The paper is written by Pickett & Carpenter. “Arthropod” is actually a part of the journal title. DOI is absent so it is difficult to correct this citation if someone is not familiar with it. Moreover, this is the same paper as cited above (no. 8). I guess other references should be checked as well.

Round 2

Reviewer 2 Report

Comments and Suggestions for Authors

It looks like the authors properly revised the manuscript taking into account all suggestions. I guess that the paper can be published now.

Author Response

We thank the reviewer for the positive assessment and for confirming that all suggestions have been properly addressed.